**Data Availability Statement:** Ethical restrictions prevent the sharing of potentially sensitive data provided by study participants over the course of

# Economic evaluation of the Target-D platform to match depression management to severity prognosis in primary care: A within-trial cost-utility analysis

Yong Yi Lee[1,2,3]*, Cathrine Mihalopoulos[1,4], Mary Lou Chatterton[1], Susan L. Fletcher[5], Patty Chondros[5], Konstancja Densley[5], Elizabeth Murray[5,6], Christopher Dowrick[5,7], Amy Coe[5], Kelsey L. Hegarty[5,8], Sandra K. Davidson[5], Caroline Wachtler[5,9], Victoria J. Palmer[5], Jane M. Gunn[5]

1 Health Economics Division, School of Public Health and Preventive Medicine, Monash University, Melbourne, Victoria, Australia, 2 School of Public Health, The University of Queensland, Brisbane, Australia, 3 Queensland Centre for Mental Health Research, Brisbane, Australia, 4 Deakin Health Economics, Institute for Health Transformation, Deakin University, Geelong, Australia, 5 Department of General Practice, Melbourne Medical School, The University of Melbourne, Parkville, Australia, 6 Research Department of Primary Care and Population Health, University College London, London, United Kingdom, 7 Department of Primary Care and Mental Health, University of Liverpool, Liverpool, United Kingdom, 8 The Royal Women's Hospital, Melbourne, Australia, 9 Department of General Practice and Primary Care, Karolinska Institutet, Solna, Sweden

☯ These authors contributed equally to this work.
\* yongyi.lee@monash.edu

## Abstract

### Background

Target-D, a new person-centred e-health platform matching depression care to symptom severity prognosis (minimal/mild, moderate or severe) has demonstrated greater improvement in depressive symptoms than usual care plus attention control. The aim of this study was to evaluate the cost-effectiveness of Target-D compared to usual care from a health sector and partial societal perspective across 3-month and 12-month follow-up.

### Methods and findings

A cost-utility analysis was conducted alongside the Target-D randomised controlled trial; which involved 1,868 participants attending 14 general practices in metropolitan Melbourne, Australia. Data on costs were collected using a resource use questionnaire administered concurrently with all other outcome measures at baseline, 3-month and 12-month follow-up. Intervention costs were assessed using financial records compiled during the trial. All costs were expressed in Australian dollars (A\$) for the 2018–19 financial year. QALY outcomes were derived using the Assessment of Quality of Life-8D (AQoL-8D) questionnaire. On a per person basis, the Target-D intervention cost between \$14 (minimal/mild prognostic group) and \$676 (severe group). Health sector and societal costs were not significantly different between trial arms at both 3 and 12 months. Relative to a A\$50,000 per QALY willingness-to-pay threshold, the probability of Target-D being cost-effective under a health sector

the Target-D clinical trial (Australian New Zealand Clinical Trials Registry ACTRN12616000537459). No contingency was included in the original plain language statement for study participants to provide informed consent to any prospective sharing of their personal data, whether as part of a minimum dataset or another form. Data requests can be sent to the Office of Research Ethics and Integrity (OREI) at The University of Melbourne (HumanEthics-Enquiries@unimelb.edu.au). For all other general enquiries regarding the Target-D clinical trial, please contact the chief investigator, Prof Jane Gunn (j.gunn@unimelb.edu.au).

**Funding:** Target-D was funded by a grant from the National Health and Medical Research Council (NHMRC project ID: 1059863). The funding organisation had no role in the design and conduct of the study; collection, management analysis, and interpretation of the data; preparation, review, or approval of the manuscript; and decision to submit the manuscript for publication.

**Competing interests:** The authors have declared that no competing interests exist.

perspective was 81% at 3 months and 96% at 12 months. From a societal perspective, the probability of cost-effectiveness was 30% at 3 months and 80% at 12 months.

## Conclusions

Target-D is likely to represent good value for money for health care decision makers. Further evaluation of QALY outcomes should accompany any routine roll-out to assess comparability of results to those observed in the trial. This trial is registered with the Australian New Zealand Clinical Trials Registry (ACTRN12616000537459).

## Introduction

Depression is a leading cause of disease burden globally and is associated with widespread economic impacts [1, 2]. Several effective treatments for depression exist, including psychotherapy and pharmacotherapy [3–5]. Stepped care approaches, which start from a low-intensity intervention before transitioning to increasing levels of treatment, are recommended by several depression treatment guidelines as optimal clinical practice [3–5]. While evidence exists on the effectiveness/cost-effectiveness of stepped care approaches for anxiety disorders, there is comparatively little evidence for these approaches when treating depression [6]. Target-D is a new approach comprising a clinical prediction tool (CPT) embedded within a person-centred e-health platform that provides symptom feedback, priority setting and management options matched to the predicted severity of an individual's depressive symptoms in three months' time (i.e., minimal/mild, moderate or severe) [7]. A randomised controlled trial (RCT) recently showed that, relative to usual care, Target-D improves depressive symptoms at 3 months (as measured by the 9-item Patient Health Questionnaire [PHQ-9])–i.e., a standardised mean reduction of -0.16 (95% confidence interval: -0.26 to -0.05) [8]. This study describes the economic evaluation conducted within the Target-D RCT. It aims to determine whether systematically matching depression care to symptom severity prognosis was cost-effective when compared to usual care.

## Methods

### Intervention trial

A within-trial economic evaluation was conducted across 14 general practices in metropolitan Melbourne, Australia. Study participants were aged 18–65 years and self-reported: current depressive symptoms (scoring $\geq 2$ on the PHQ-2); stable antidepressant medication use in the past month; no antipsychotic medication use or regular psychotherapy; internet access; and English proficiency. The trial compared the Target-D intervention (feedback, priority setting and prognosis-matched management options) to usual care plus attention control (a telephone call from the research team). Management options for participants in the intervention arm included: (1) unguided internet-based cognitive behavioural therapy (iCBT) (minimal/mild prognostic group) [9]; (2) clinician-guided iCBT (moderate group) [10]; and (3) nurse-led collaborative care (severe group) [11]. S1 Text (see S1 Appendix) outlines treatments received in each trial arm. The study protocol and analysis of clinical outcomes are described elsewhere [7, 8]. The authors assert that all procedures contributing to this work comply with the ethical standards of the relevant national and institutional committees on human experimentation and with the Helsinki Declaration of 1975, as revised in 2008. All procedures involving human

subjects/patients were approved by The University of Melbourne Health Sciences Human Ethics Sub-Committee (approval number 1543648).

## Cost analysis

The economic evaluation primarily adopted an Australian health sector perspective. An additional analysis was also conducted from a partial societal perspective, as recommended by recent economic evaluation guidelines [12]. Health sector costs included the cost of delivering Target-D alongside the cost of other health care services incurred by study participants over the trial period. Societal costs included all health sector costs plus the cost of self-reported productivity losses. S2 Table (see S1 Appendix) presents an impact inventory with details of each cost component by study perspective. All costs are presented in Australian dollars (A$) for the 2018–19 financial year. Discounting was not required.

A micro-costing approach was used to estimate the cost of the Target-D intervention based on trial data (see S3 Table in S1 Appendix). Intervention costs were divided into: (1) the costs of screening using the Target-D CPT; and (2) the costs of treating participants within each assigned prognostic group. Screening costs were sourced from the study team which comprised the cost of hardware (iPads and Wi-Fi dongles) and the cost of CPT maintenance. Personnel time to approach individuals in the GP waiting room involved one minute per encounter and was costed using the relevant wage rate (plus overheads) for a research assistant. The average cost per screened person was computed and applied to everyone in the intervention arm.

Intervention costs for participants in the minimal/mild prognostic group comprised the annual registration cost of an Australian unguided iCBT program [9] and the cost of research assistants providing an initial check-in phone call. The average cost between two Australian clinician-guided iCBT programs was applied to participants in the moderate prognostic group [10, 13], alongside the cost of research assistants providing periodic check-in phone calls to monitor participant progress. In this instance, the two programs represent a low-to-high range of possible unit cost values for clinician-guided iCBT in Australia. A subsequent sensitivity analysis was done to test the impact of using the highest unit cost, rather than the average. Intervention costs for participants in the severe prognostic group encompassed time/resources incurred by nurses to administer collaborative care (laptops, phones, etc). It is anticipated that all costed activities described above that involve research assistants will likely require similarly qualified staff to facilitate the implementation of the intervention as part of routine practice.

Participants were asked to complete a resource use questionnaire (RUQ) at baseline, 3-month and 12-month follow-up. The RUQ measured: health care professional visits (GPs, psychologists, psychiatrists, etc.); diagnostic tests; medication usage; hospitalisations; emergency department visits; community mental health contacts; and time taken off from paid and unpaid work (see S4 Text in S1 Appendix).

## Health outcomes

The Assessment of Quality of Life-8D (AQoL-8D) was used to measure participants' health-related quality of life at baseline, 3-month and 12-month follow-up [14]. Australian general population preference weights were applied to calculate utility weights at each time point [15]. Utility weights were then used to estimate quality-adjusted life years (QALYs) based on the area-under-the-curve (AUC) method [16].

## Statistical analysis

Data management was implemented using Excel 2016 and R version 3.6.0. Statistical analyses were conducted using Stata 15 (College Station, Texas, USA). Researchers undertaking the

cost analysis were not blinded to trial arm allocation. Base case analyses were conducted as intention-to-treat from the health sector and societal perspectives. All enrolled participants who completed a baseline assessment were included. However, 66% participants were missing RUQ data at 3 months (46%) or 12 months (60%); while 68% were missing AQoL-8D data at 3 months (49%) or 12 months (66%). Multiple imputation methods were implemented in Stata to account for missing data that were deemed missing at random following several exploratory analyses presented in S5 Text in S1 Appendix. Missing cost and outcomes data were imputed 100 times using multiple imputation by chained equations (MICE), with predictive mean matching and adjustment for baseline covariates associated with data missingness–i.e., trial arm, clinic, age, gender, highest level of education and having visited a psychologist/counsellor in the past 12 months.

The difference in mean QALYs between trial arms was estimated using a generalised linear model (GLM) involving the 'Gaussian' family and 'identity' link. The ratio of mean costs from both health sector and societal perspectives were estimated using GLMs involving the 'gamma' family and 'log' link. All GLMs were estimated with and without adjustment for several baseline covariates specified in the study protocol – i.e., baseline PHQ-9 score, general practice and prognostic group [7]. Baseline AQoL-8D utility weights were also included as an additional baseline covariate for GLMs involving QALY outcomes. Subgroup analyses were conducted across the three prognostic groups.

Incremental cost-effectiveness ratios (ICERs) were calculated as the difference in mean costs between the intervention and control arms divided by the difference in mean QALYs. ICERs were calculated by study perspective (health sector and societal), follow-up period (3 and 12 months) and, for the subgroup analysis, by prognostic group (total, minimal/mild, moderate and severe). A resampling method comprising single imputation nested in boot-strapping [17] was used to quantify the impact of input parameter uncertainty around the resulting differences in mean costs/QALYs and the mean ICERs. This method works by generating a single call to the MICE procedure to produce a complete dataset with which to analyse GLMs of costs/QALYs within each bootstrap resample. Following the generation of 1,000 bootstrap resamples, the bootstrap percentile method was used to estimate 95% confidence intervals (95% CI) around the differences in mean costs/QALYs and the mean ICERs [18]. The intervention was considered cost-effective if the resulting ICER was less than the Australian willingness-to-pay threshold of A$50,000 per QALY [19–21].

A summary list of all variables included in the statistical analysis is provided in S2 Appendix, alongside the Stata do-file used to implement the statistical analysis.

## Sensitivity analysis

Sensitivity analysis was conducted to examine the impact of including additional baseline covariates associated with non-response at 3 and 12 months that were identified in the primary outcomes analysis (i.e., age, gender, highest level of education, current employment status, health care card status, long term illness, living alone, number of times visited a psychiatrist or counsellor in past 12 months and current use of antidepressants) [8]. Two additional sensitivity analyses investigated the impact of changes to the intervention costing. The first involved increasing the unit cost for the clinician-guided iCBT program delivered to participants in the moderate prognostic group to the highest unit cost among the two alternative programs (average cost changed from $132 to $222 per person). The second involved incorporating sunk costs around Target-D CPT research and development (average cost of screening changed from $0.96 to $2.30 per person). Additionally, a complete case analysis was conducted for QALYs and costs to examine the impact of no multiple imputation of missing data.

# Results

## Intervention trial

Most of the 1,868 trial participants were allocated to the minimal/mild prognostic group (72.6%, n = 1,357, intervention = 679, control = 678); with 288 (15.4%, intervention = 143, control = 145) to the moderate group; and 223 (11.9%, intervention = 111, control = 112) to the severe group. Participants across both trial arms were similar overall and within prognostic groups [8].

## Cost analysis

The cost of screening was estimated to be $0.96 per person after excluding sunk costs related to Target-D CPT research and development (see S3 Table in S1 Appendix). The cost of Target-D treatment in the intervention arm was: $14 per person for the minimal/mild prognostic group; $132 per person for the moderate group; and $676 per person in the severe group.

Table 1 shows the estimated ratio of mean health sector costs between the intervention and control arms. Health sector costs were comparable between trial arms at 3-month follow-up.

**Table 1. Comparison of health sector costs by trial arm, across all participants and stratified by prognostic group (multiple imputed data).**

|  | All participants (n = 1,868) | p-value | Minimal/mild (n = 1,357) | p-value | Moderate (n = 288) | p-value | Severe (n = 223) | p-value |
|---|---|---|---|---|---|---|---|---|
| **3 months** |  |  |  |  |  |  |  |  |
| Mean costs (SE) [1] |  |  |  |  |  |  |  |  |
| *Intervention arm* | 625 (54) |  | 487 (52) |  | 812 (138) |  | 1,842 (265) |  |
| *Control arm* | 630 (52) |  | 509 (50) |  | 947 (148) |  | 966 (148) |  |
| Ratio of mean costs between arms (95% CI) [1] | 0.99 (0.78 to 1.25) | 0.94 | 0.96 (0.72 to 1.27) | 0.76 | 0.86 (0.55 to 1.33) | 0.49 | 1.91 (1.24 to 2.93) | 0.003 |
| Sensitivity analysis [2] | 1.04 (0.83 to 1.31) | 0.72 | 1.02 (0.77 to 1.35) | 0.88 | 0.96 (0.65 to 1.42) | 0.85 | 1.93 (1.34 to 2.79) | 0.001 |
| Sensitivity analysis [3] | 1.01 (0.80 to 1.27) | 0.95 | 0.96 (0.72 to 1.27) | 0.76 | 0.97 (0.63 to 1.48) | 0.87 | 1.91 (1.24 to 2.92) | 0.003 |
| Sensitivity analysis [4] | 0.99 (0.79 to 1.25) | 0.96 | 0.96 (0.72 to 1.27) | 0.77 | 0.86 (0.55 to 1.33) | 0.50 | 1.91 (1.24 to 2.93) | 0.003 |
| Sensitivity analysis [5] | 0.96 (0.75 to 1.24) | 0.77 | 0.96 (0.71 to 1.29) | 0.77 | 0.74 (0.45 to 1.24) | 0.26 | 1.72 (1.01 to 2.92) | 0.046 |
| **12 months** |  |  |  |  |  |  |  |  |
| Mean costs (SE) [1] |  |  |  |  |  |  |  |  |
| *Intervention arm* | 1,643 (112) |  | 1,389 (122) |  | 1,864 (237) |  | 3,517 (422) |  |
| *Control arm* | 1,798 (115) |  | 1,565 (127) |  | 2,134 (246) |  | 2,632 (346) |  |
| Ratio of mean costs between arms (95% CI) [1] | 0.91 (0.76 to 1.09) | 0.32 | 0.89 (0.70 to 1.12) | 0.31 | 0.87 (0.63 to 1.21) | 0.42 | 1.34 (0.94 to 1.90) | 0.11 |
| Sensitivity analysis [2] | 0.94 (0.78 to 1.13) | 0.50 | 0.91 (0.72 to 1.15) | 0.45 | 0.89 (0.66 to 1.21) | 0.47 | 1.33 (0.95 to 1.86) | 0.09 |
| Sensitivity analysis [3] | 0.92 (0.77 to 1.10) | 0.36 | 0.89 (0.70 to 1.12) | 0.31 | 0.92 (0.67 to 1.27) | 0.61 | 1.34 (0.94 to 1.90) | 0.11 |
| Sensitivity analysis [4] | 0.91 (0.76 to 1.09) | 0.33 | 0.89 (0.70 to 1.12) | 0.31 | 0.87 (0.63 to 1.21) | 0.42 | 1.34 (0.94 to 1.90) | 0.11 |
| Sensitivity analysis [5] | 0.87 (0.69 to 1.10) | 0.25 | 0.89 (0.67 to 1.18) | 0.42 | 0.66 (0.42 to 1.06) | 0.09 | 1.16 (0.62 to 2.15) | 0.65 |

Abbreviations: SE = standard error; CI = confidence interval

[1] Baseline mean and the ratio of the mean for the intervention arm and control arm estimated using a generalised linear model (family = gamma, link = log) with random intercepts for individuals and adjusted for baseline PHQ-9 score, general practice and prognostic group (the final covariate only applied to the analysis involving all participants);

[2] Same as 1, adjusted for factors associated with non-response to the primary outcome measure, the PHQ-9 score, at 3 and 12 months (age, gender, highest level of education, current employment status, hold a health care card, long-term illness, live alone, number of times visited a psychiatrist or counsellor in past 12 months and current use of antidepressants);

[3] Same as 1, but using a higher unit cost for the clinician-guided iCBT course delivered to the moderate prognostic group (unit cost changed from $132 per person to $222 per person);

[4] Same as 1, but with the inclusion of sunk costs for the development of the Target-D CPT (cost of screening changed from $0.96 per person to $2.30 per person);

[5] Same as 1, for complete cases only (i.e., no multiple imputation of missing data)

No statistically significant differences were detected across all participants or the minimal/mild and moderate prognostic groups. However, mean health sector costs were 1.9 times higher (*95% CI: 1.24 to 2.93*) among the intervention arm relative to the control arm for those in the severe prognostic group. This was likely due to the high-cost nature of collaborative care delivered to participants in the severe group (see S3 Table in S1 Appendix for detailed costs). At 12 months, no statistically significant differences were observed overall or across each of the prognostic groups. Health sector costs were 9% lower (not significant) in the intervention arm relative to the control arm (*ratio of 0.91, 95% CI: 0.76 to 1.09*). Lower health sector costs were also observed in the minimal/mild and moderate prognostic groups (*ratios of 0.89 and 0.87, respectively*). In the severe prognostic group, health sector costs were 34% higher in the intervention arm at 12-month follow-up (*95% CI: -6% to +90%*). The estimated ratio of mean societal costs between the intervention and control arms are presented in Table 2. Mean costs were higher (but not significant) in the intervention arm at 3-month follow-up for all participants and across each of the three prognostic groups.

**Table 2. Comparison of societal costs by trial arm, across all participants and stratified by prognostic group (multiple imputed data).**

| | All participants (n = 1,868) | p-value | Minimal/mild (n = 1,357) | p-value | Moderate (n = 288) | p-value | Severe (n = 223) | p-value |
|---|---|---|---|---|---|---|---|---|
| **3 months** | | | | | | | | |
| Mean costs (SE) [1] | | | | | | | | |
| *Intervention arm* | 5,326 (244) | | 5,093 (277) | | 5,462 (704) | | 6,133 (794) | |
| *Control arm* | 4,966 (225) | | 4,856 (258) | | 5,296 (614) | | 4,537 (663) | |
| Ratio of mean costs between arms (95% CI)[1] | 1.07 (0.95 to 1.21) | 0.27 | 1.05 (0.90 to 1.22) | 0.53 | 1.03 (0.73 to 1.46) | 0.86 | 1.35 (0.90 to 2.04) | 0.15 |
| Sensitivity analysis[2] | 1.07 (0.94 to 1.23) | 0.30 | 1.05 (0.89 to 1.24) | 0.53 | 1.01 (0.69 to 1.48) | 0.94 | 1.31 (0.87 to 2.02) | 0.23 |
| Sensitivity analysis[3] | 1.07 (0.95 to 1.22) | 0.25 | 1.05 (0.90 to 1.22) | 0.53 | 1.05 (0.75 to 1.48) | 0.78 | 1.35 (0.90 to 2.04) | 0.15 |
| Sensitivity analysis[4] | 1.07 (0.95 to 1.21) | 0.27 | 1.05 (0.90 to 1.22) | 0.53 | 1.03 (0.73 to 1.46) | 0.86 | 1.35 (0.90 to 2.04) | 0.15 |
| Sensitivity analysis[5] | 1.08 (0.94 to 1.24) | 0.31 | 1.05 (0.90 to 1.23) | 0.54 | 1.11 (0.74 to 1.67) | 0.60 | 1.28 (0.84 to 1.96) | 0.25 |
| **12 months** | | | | | | | | |
| Mean costs (SE) [1] | | | | | | | | |
| *Intervention arm* | 17,159 (662) | | 17,053 (796) | | 16,104 (1,542) | | 17,978 (2,161) | |
| *Control arm* | 17,538 (632) | | 17,316 (769) | | 18,246 (1,511) | | 16,731 (2,013) | |
| Ratio of mean costs between arms (95% CI)[1] | 0.98 (0.88 to 1.08) | 0.67 | 0.98 (0.87 to 1.11) | 0.81 | 0.88 (0.68 to 1.14) | 0.34 | 1.07 (0.76 to 1.52) | 0.67 |
| Sensitivity analysis[2] | 0.98 (0.88 to 1.09) | 0.71 | 0.97 (0.85 to 1.10) | 0.62 | 0.85 (0.64 to 1.12) | 0.25 | 1.03 (0.70 to 1.52) | 0.86 |
| Sensitivity analysis[3] | 0.98 (0.89 to 1.08) | 0.68 | 0.98 (0.87 to 1.11) | 0.81 | 0.89 (0.69 to 1.15) | 0.36 | 1.07 (0.76 to 1.52) | 0.69 |
| Sensitivity analysis[4] | 0.98 (0.88 to 1.08) | 0.67 | 0.98 (0.87 to 1.11) | 0.81 | 0.88 (0.68 to 1.14) | 0.34 | 1.07 (0.76 to 1.52) | 0.69 |
| Sensitivity analysis[5] | 0.94 (0.81 to 1.08) | 0.36 | 1.00 (0.85 to 1.18) | 0.96 | 0.79 (0.51 to 1.20) | 0.27 | 0.75 (0.46 to 1.20) | 0.23 |

Abbreviations: SE = standard error; CI = confidence interval

[1] Baseline mean and the ratio of the mean for the intervention arm and control arm estimated using a generalised linear model (family = gamma, link = log) with random intercepts for individuals and adjusted for baseline PHQ-9 score, general practice and prognostic group (the final covariate only applied to the analysis involving all participants);

[2] Same as 1, adjusted for factors associated with non-response to the primary outcome measure, the PHQ-9 score, at 3 and 12 months (age, gender, highest level of education, current employment status, hold a health care card, long-term illness, live alone, number of times visited a psychiatrist or counsellor in past 12 months and current use of antidepressants);

[3] Same as 1, but using a higher unit cost for the clinician-guided iCBT course delivered to the moderate prognostic group (unit cost changed from $132 per person to $222 per person);

[4] Same as 1, but with the inclusion of sunk costs for the development of the Target-D CPT (cost of screening changed from $0.96 per person to $2.30 per person)

[5] Same as 1, for complete cases only (i.e., no multiple imputation of missing data)

## Health outcomes

Estimated differences in mean QALYs resulted in no significant differences between trial arms at 3 months (Table 3), although there was a trend towards marginally higher QALYs in the intervention arm overall (*difference of 0.002 across all participants*, *95% CI: -0.0002 to 0.003*) (Table 3) and in the moderate prognostic group (*0.004, 95% CI: -0.001 to 0.008*). At 12 months, the difference in mean QALYs was significant overall (*0.011, 95% CI: 0.001 to 0.022*), but not within prognostic groups. This was possibly due to insufficient power in the subgroup analysis.

## Cost-effectiveness results

Cost-effectiveness results are presented in Table 4. When adopting a health sector perspective, the intervention was found to be dominant across all participants at both 3 and 12 months – i.e., the intervention jointly improved health and achieved net cost savings. Under the societal perspective, the intervention was only found to be dominant across all participants at 12 months. The ICER at 3 months exceeded the Australian willingness-to-pay threshold of A\$50,000 per QALY (*$237,128, 95% CI: dominant to dominated*). Cost-effectiveness planes and cost-effectiveness acceptability curves are presented in: S6 and S7 Figs for results at 3-month

**Table 3. Comparison of quality-adjusted life years by trial arm, across all participants and stratified by prognostic group (multiple imputed data).**

|  | All participants (n = 1,868) | p-value | Minimal/mild (n = 1,357) | p-value | Moderate (n = 288) | p-value | Severe (n = 223) | p-value |
|---|---|---|---|---|---|---|---|---|
| **3 months** |  |  |  |  |  |  |  |  |
| Mean QALYs (SE) [1] |  |  |  |  |  |  |  |  |
| *Intervention arm* | 0.147 (0.0006) |  | 0.164 (0.0008) |  | 0.112 (0.002) |  | 0.089 (0.002) |  |
| *Control arm* | 0.145 (0.0006) |  | 0.163 (0.0008) |  | 0.108 (0.001) |  | 0.085 (0.002) |  |
| Difference in mean QALYs between arms (95% CI)[1] | 0.002 (-0.0002 to 0.003) | 0.09 | 0.0008 (-0.001 to 0.003) | 0.47 | 0.004 (-0.001 to 0.008) | 0.13 | 0.003 (-0.002 to 0.008) | 0.22 |
| Sensitivity analysis[2] | 0.001 (-0.0004 to 0.003) | 0.15 | 0.0007 (-0.001 to 0.003) | 0.52 | 0.003 (-0.001 to 0.008) | 0.16 | 0.003 (-0.002 to 0.008) | 0.19 |
| Sensitivity analysis[3] | 0.001 (-0.0009 to 0.003) | 0.27 | -0.0002 (-0.003 to 0.002) | 0.85 | 0.004 (-0.0008 to 0.009) | 0.10 | 0.004 (-0.001 to 0.010) | 0.17 |
| **12 months** |  |  |  |  |  |  |  |  |
| Mean QALYs (SE) [1] |  |  |  |  |  |  |  |  |
| *Intervention arm* | 0.607 (0.004) |  | 0.669 (0.005) |  | 0.482 (0.011) |  | 0.392 (0.012) |  |
| *Control arm* | 0.596 (0.004) |  | 0.661 (0.004) |  | 0.465 (0.010) |  | 0.367 (0.011) |  |
| Difference in mean QALYs between arms (95% CI)[1] | 0.011 (0.001 to 0.022) | 0.049 | 0.008 (-0.005 to 0.021) | 0.23 | 0.016 (-0.014 to 0.047) | 0.29 | 0.024 (-0.011 to 0.059) | 0.18 |
| Sensitivity analysis[2] | 0.010 (-0.001 to 0.021) | 0.09 | 0.00 (-0.006 to 0.020) | 0.29 | 0.016 (-0.015 to 0.046) | 0.32 | 0.024 (-0.010 to 0.059) | 0.16 |
| Sensitivity analysis[3] | 0.015 (0.00004 to 0.030) | 0.049 | 0.009 (-0.008 to 0.027) | 0.32 | 0.048 (0.002 to 0.095) | 0.04 | 0.013 (-0.034 to 0.059) | 0.59 |

Abbreviations: QALYs = quality-adjusted life years; SE = standard error; CI = confidence interval

[1] Baseline mean and the difference between the mean for the intervention arm minus the mean for control arm estimated using a generalised linear model (family = Gaussian, link = identity) with random intercepts for individuals and adjusted for baseline AQoL-8D utility weight, baseline PHQ-9 score, general practice and prognostic group (the final covariate only applied to the analysis involving all participants);

[2] Same as 1, adjusted for factors associated with non-response to the primary outcome measure, the PHQ-9 score, at 3 and 12 months (age, gender, highest level of education, current employment status, hold a health care card, long-term illness, live alone, number of times visited a psychiatrist or counsellor in past 12 months and current use of antidepressants)

[3] Same as 1, for complete cases only (i.e., no multiple imputation of missing data)

**Table 4. Incremental cost-effectiveness ratios under the health sector and societal perspectives, across all participants and stratified by prognostic group.**

| | All participants (n = 1,868) | Minimal/mild (n = 1,357) | Moderate (n = 288) | Severe (n = 223) |
|---|---|---|---|---|
| **Health sector perspective** | | | | |
| ICER (95% CI)[1] | | | | |
| *3 months* | Dominant[2] (Dominant[2] to Dominated[3]) | Dominant[2] (Dominant[2] to Dominated[3]) | Dominant[2] (Dominant[2] to Dominated[3]) | 364,966 (91,062 to Dominated[3]) |
| *12 months* | Dominant[2] (Dominant[2] to 87,636) | Dominant[2] (Dominant[2] to Dominated[3]) | Dominant[2] (Dominant[2] to Dominated[3]) | 45,424 (Dominant[2] to Dominated[3]) |
| **Societal perspective** | | | | |
| ICER (95% CI)[1] | | | | |
| *3 months* | 237,128 (Dominant[2] to Dominated[3]) | 254,155 (Dominant[2] to Dominated[3]) | 52,762 (Dominant[2] to Dominated[3]) | 721,711 (Dominant[2] to Dominated[3]) |
| *12 months* | Dominant[2] (Dominant[2] to 140,191) | Dominant[2] (Dominant[2] to Dominated[3]) | Dominant[2] (Dominant[2] to Dominated[3]) | 57,039 Dominant[2] to Dominated[3]) |

Abbreviations: CI = confidence interval; ICER = incremental cost-effectiveness ratio

[1] Mean incremental costs and mean incremental QALYs estimated using a generalised linear model (family = gamma, link = log) with random intercepts for individuals and adjusted for baseline AQoL-8D utility weight (mean incremental QALYs only), baseline PHQ-9 score, general practice and prognostic group (the final covariate only applied to the analysis involving all participants). Confidence intervals were estimated for the mean ICER based on 1,000 bootstrap resamples. The mean difference in QALYs between trial arms was observed to approach zero across all base case and subgroup analyses. This can potentially lead to the lower and upper bounds of a 95% confidence interval, derived using the bootstrap percentile method, encompassing a marginally higher coverage than the target 95% confidence region (e.g., 97% coverage of the mean ICER) [22]. Even so, any resulting imprecision in the estimation of the 95% confidence bounds will likely be inconsequential to the interpretation of study findings given the wide range of ICER values that were consistently observed between the lower and upper confidence bounds (e.g., confidence bounds ranging between 'dominant' and 'dominated'). This reflects the high degree of uncertainty observed across mean ICER values for all base case and subgroup analyses; with bootstrap resamples consistently spanning all four quadrants of the cost-effectiveness plane.

[2] A 'dominant' ICER indicates that the intervention costs less and is more effective than the control.

[3] A 'dominated' ICER indicates that the intervention costs more and is less effective than the control.

follow-up (see S1 Appendix); and Figs 1 and 2 at 12-month follow-up. The probability of the intervention being cost-effective when adopting the health sector perspective was 81% at 3 months (S6 Fig in S1 Appendix) and 96% at 12 months (Fig 1). From the societal perspective, the probability of the intervention being cost-effective was 30% at 3 months (S7 Fig in S1 Appendix) and 80% at 12 months (Fig 2).

In the subgroup analysis, the intervention was dominant when analysing the minimal/mild and moderate prognostic groups at 3 and 12 months under the health sector perspective (see Table 4). The ICER for the severe group exceeded the willingness-to-pay threshold at 3 months (*$364,966, 95% CI: $91,062 to dominated*), then fell below this threshold at 12 months (*$45,424, 95% CI: dominant to dominated*). When adopting the societal perspective, at 3-months the ICERs for the minimal/mild, moderate and severe groups exceeded the willingness-to-pay threshold. At 12 months, however, the ICERs under the societal perspective were observed to be: dominant among the minimal/mild and moderate groups; and just above the willingness-to-pay threshold among the severe group.

## Sensitivity analysis

Mean health sector costs and mean societal costs between trial arms were robust to sensitivity analysis (Tables 1 and 2). The addition of covariates associated with non-response (Table 3) reduced the 12-month difference in mean QALYs across all participants, which ceased to be significant (*0.010, 95% CI: -0.001 to 0.021*). In the complete case analysis (Table 3), the 12-month difference in mean QALYs remained significant across all participants (*0.015, 95%*

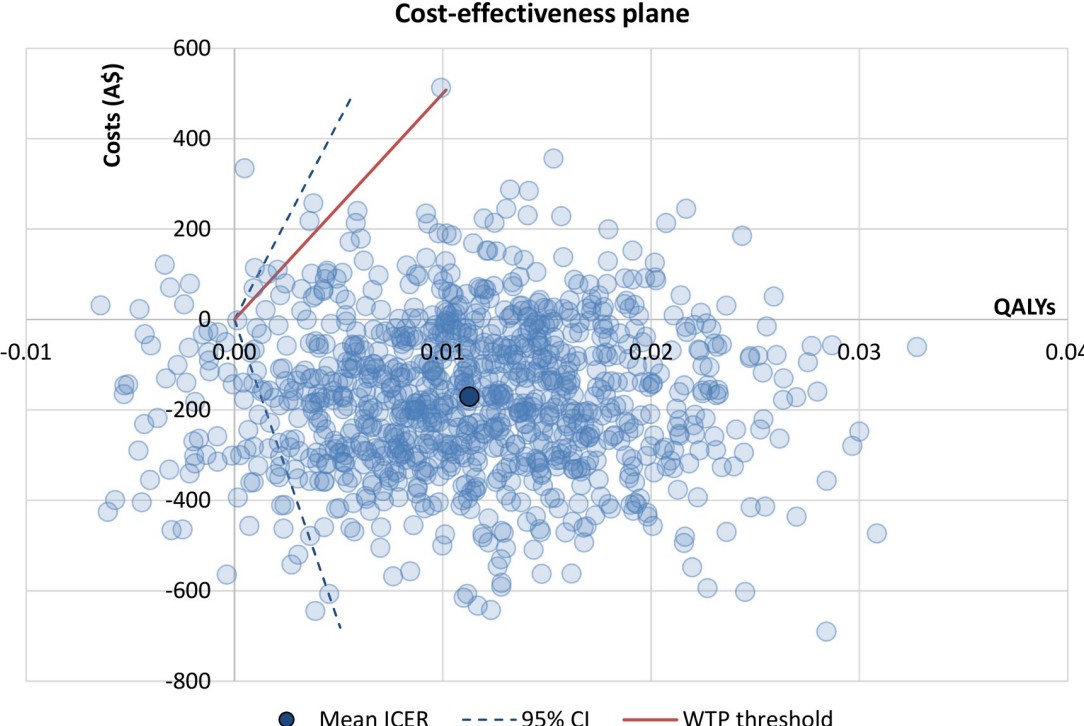

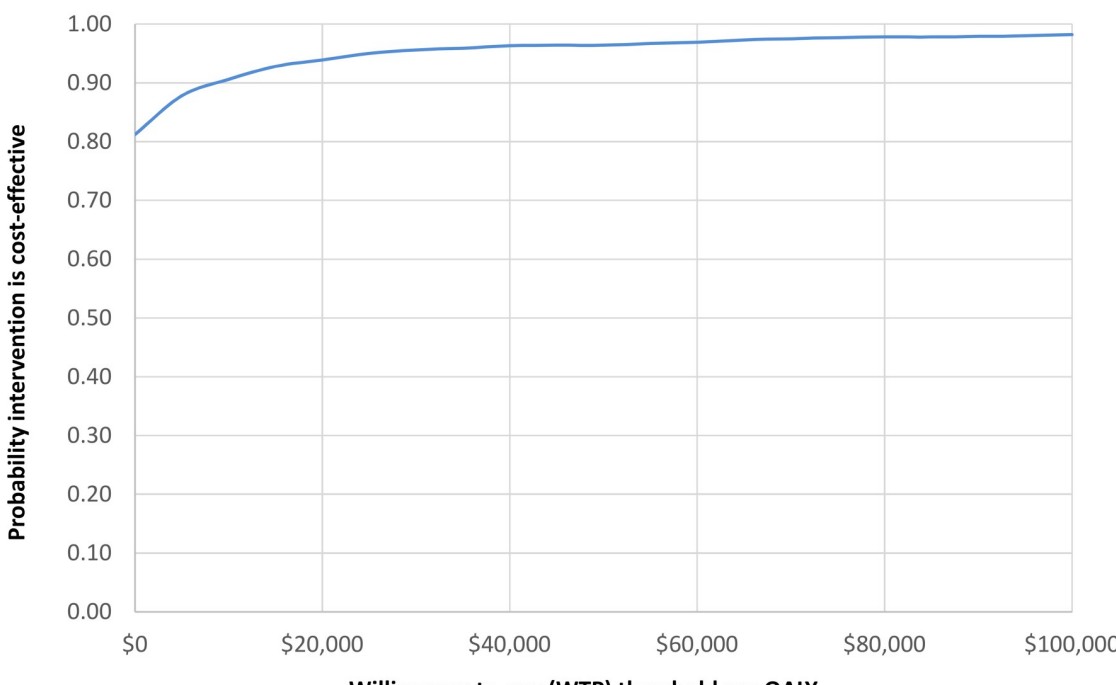

**Fig 1. Cost-effectiveness results for the health sector perspective across all participants at 12 months.** Abbreviations: A$ = Australian dollars; CI = confidence interval; ICER = incremental cost-effectiveness ratio; QALYs = quality-adjusted life years; WTP = willingness to pay.

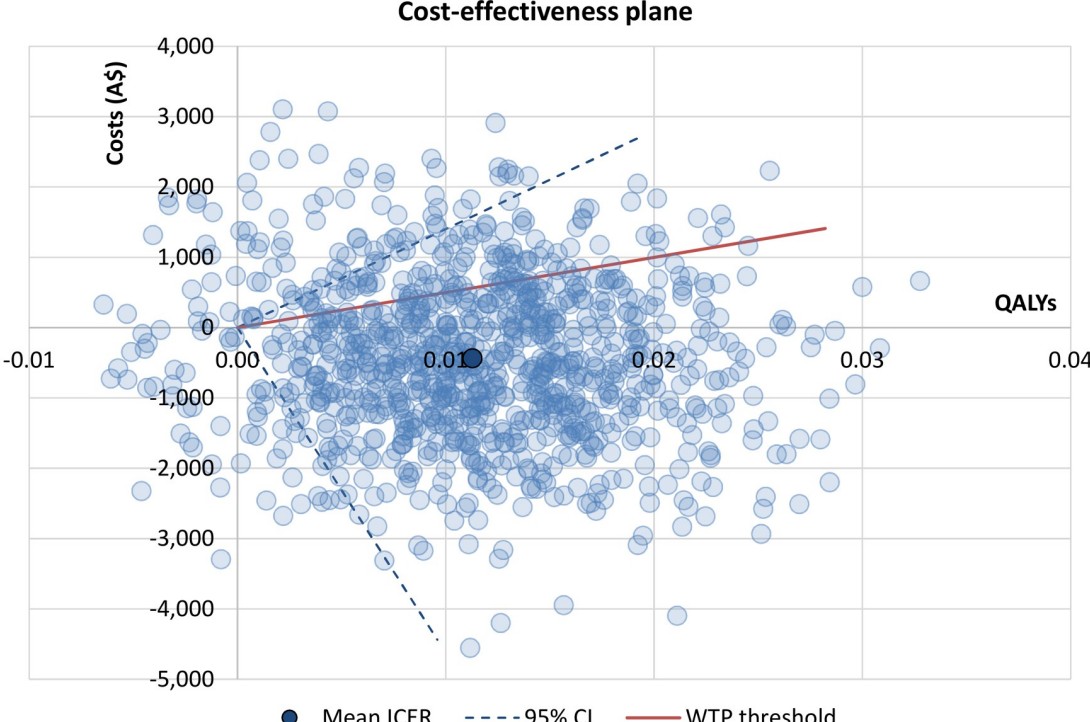

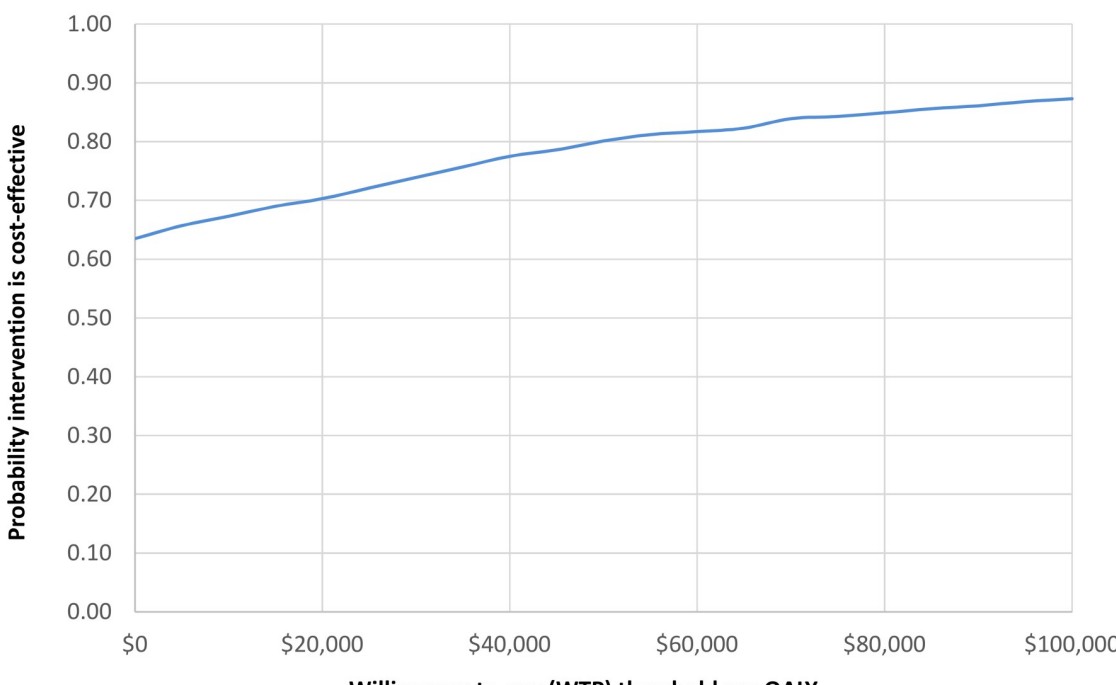

**Fig 2. Cost-effectiveness results for the societal perspective across all participants at 12 months.** Abbreviations: A$ = Australian dollars; CI = confidence interval; ICER = incremental cost-effectiveness ratio; QALYs = quality-adjusted life years; WTP = willingness to pay.

*CI*: *0.00004 to 0.030*). The 12-month difference in mean QALYs became significant in the moderate prognostic group (*0.048, 95% CI*: *0.002 to 0.095*). This could either be a chance finding or an indication that the missing-at-random assumption underlying multiple imputation does not hold; though it is highly unlikely data are missing completely at random (as is assumed in complete case analysis).

## Discussion

### Summary

This is the first economic evaluation of an e-health platform designed to personally tailor depression management in primary care. The results suggest that over 12 months, the screening and intervention program is likely to be a more effective and less costly approach than usual GP care. This finding is driven by an observed difference in quality of life between the two trial arms and no significant differences in costs at 12-month follow-up.

### Strengths and limitations

Major strengths of this study include the large sample size and the incorporation of economic data collection within the RCT design, thus adding greater certainty to results. The study also used a quality of life instrument that is sensitive to measuring health states among people with mental health problems, thus increasing the likelihood that any quality of life impacts resultant from the intervention would be detected [14, 23]. Use of a self-reported measure of health care utilisation may be a limitation due to recall bias; though previous studies have demonstrated the validity of self-reported resource use questionnaires [24]. While more frequent data collection (e.g., every three months) may minimise recall bias, the additional burden it places on participants and trial staff alike was not considered appropriate for this trial. The resource use questionnaire was thoroughly piloted prior to the commencement of the study and the study authors are confident that all main resource use categories relevant to this population were included. Sensitivity analyses demonstrated that QALY gains became non-significant after incorporating covariates for missing data involving the primary outcome (i.e., PHQ-9 scores). This suggests that while significant differences in QALYs were observed at 12 months in the base case analysis, these differences may be considered uncertain; particularly given that differences were not observed for the primary outcome of the main RCT [8]. The study sample involved an urban population in Melbourne which may limit the generalisability of study findings to other settings (e.g., other countries or rural regions of Australia). Study findings are limited by high levels of missing RUQ and AQoL-8D data, a common issue in trial-based economic evaluation [25]. Finally, the study was powered for the primary outcome only and may be underpowered to detect differences in costs or QALYs [7, 8].

### Comparison with existing literature

Interventions that are directly comparable to the Target-D approach have not been evaluated in the literature to date. However, Grochtdreis, Brettschneider [26] reviewed several cost-effectiveness studies of collaborative care interventions for depression and which share common features with the Target-D intervention. This review identified ICERs ranging from dominant to US$874,562 per QALY and concluded that future research should incorporate a time horizon of one year or more and QALYs as an outcome measure–both of which were adopted in the current study. Another recent review of the cost-effectiveness of stepped care interventions for depression and anxiety identified one cost-effectiveness study of stepped care targeting visually impaired older people with both depression and anxiety [27]. The results of this study

suggest that stepped care may be dominant when compared to usual care. Additionally, study findings appear to support the existing literature by suggesting that stepped care for depression can deliver improved clinical outcomes without increasing costs.

## Conclusion

The Target-D intervention is likely to represent good value for money and provides indicative support for further development of digitally supported mental health care. Refinement and further testing of the approach is warranted to determine whether the size and extent of QALY gains at 12 months can be replicated. It remains to be seen whether results observed under tightly controlled trial conditions will still occur under routine service delivery conditions. In particular, the routine roll-out of the intervention needs to consider how screening will occur (e.g., by staff at general practices or online).

## Supporting information

**S1 Appendix. Supplementary materials.**
(PDF)

**S2 Appendix. Variable list and Stata do-file.**
(PDF)

## Acknowledgments

The authors would like to thank all the patients, family physicians, and clinics who took part in Target-D; and the many research assistants who assisted with data collection. The data used to develop the clinical prediction tool were collected as a part of the diamond project (NHMRC project ID: 299869, 454463, 566511 and 1002908). We acknowledge the 30 dedicated family physicians, their patients, and clinic staff for making the diamond study possible. We also acknowledge staff and students at the School of Computing and Information Systems at the University of Melbourne for early work that informed the presentation of the e-health platform as well as the focus group participants that provided feedback on early versions of the Target-D materials. Finally, we thank staff at the former Melbourne Networked Society Institute (MNSI) who built the Target-D website.

## Author Contributions

**Conceptualization:** Cathrine Mihalopoulos.

**Data curation:** Yong Yi Lee, Mary Lou Chatterton.

**Formal analysis:** Yong Yi Lee, Cathrine Mihalopoulos, Mary Lou Chatterton.

**Funding acquisition:** Cathrine Mihalopoulos, Patty Chondros, Kelsey L. Hegarty, Sandra K. Davidson, Jane M. Gunn.

**Investigation:** Yong Yi Lee, Cathrine Mihalopoulos, Mary Lou Chatterton.

**Methodology:** Yong Yi Lee, Cathrine Mihalopoulos, Mary Lou Chatterton.

**Project administration:** Susan L. Fletcher.

**Validation:** Yong Yi Lee, Cathrine Mihalopoulos, Mary Lou Chatterton.

**Writing – original draft:** Yong Yi Lee, Cathrine Mihalopoulos, Mary Lou Chatterton, Susan L. Fletcher, Patty Chondros.

**Writing – review & editing:** Yong Yi Lee, Cathrine Mihalopoulos, Mary Lou Chatterton, Susan L. Fletcher, Patty Chondros, Konstancja Densley, Elizabeth Murray, Christopher Dowrick, Amy Coe, Kelsey L. Hegarty, Sandra K. Davidson, Caroline Wachtler, Victoria J. Palmer, Jane M. Gunn.

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
