## [Decision Letter · Decision Letter 0]

22 Oct 2021

PONE-D-21-25251Economic evaluation of the Target-D platform to match depression management to severity prognosis in primary care: a within-trial cost-utility analysisPLOS ONE

Dear Dr. Lee,

Thank you for submitting your manuscript to PLOS ONE. After careful consideration, we feel that it has merit but does not fully meet PLOS ONE’s publication criteria as it currently stands. Therefore, we invite you to submit a revised version of the manuscript that addresses all the points raised during the review process, with particular attention given to the econmic comments of reviwer 1 .

We look forward to receiving your revised manuscript.

Kind regards,

Isabelle Durand-Zaleski

Academic Editor

PLOS ONE

“The authors would like to thank all the patients, family physicians, and clinics who took part

in Target-D; and the many research assistants who assisted with data collection. The data used

to develop the clinical prediction tool were collected as a part of the diamond project which

was funded by the National Health and Medical Research Council (NHMRC project ID:

299869, 454463, 566511 and 1002908). We acknowledge the 30 dedicated family physicians,

their patients, and clinic staff for making the diamond study possible. We also acknowledge

staff and students at the School of Computing and Information Systems at the University of

Melbourne for early work that informed the presentation of the e-health platform as well as the

focus group participants that provided feedback on early versions of the Target-D materials.

Finally, we thank staff at the former Melbourne Networked Society Institute (MNSI) who were

funded to build the Target-D website.”

“Target-D was funded by a grant from the National Health and Medical Research Council (NHMRC project ID: 1059863). The funding organisation had no role in the design and conduct of the study; collection, management analysis, and interpretation of the data; preparation, review, or approval of the manuscript; and decision to submit the manuscript for publication.”

Reviewers' comments:

Reviewer's Responses to Questions

**Comments to the Author**

1. Is the manuscript technically sound, and do the data support the conclusions?

Reviewer #1: Yes

Reviewer #2: Yes

2. Has the statistical analysis been performed appropriately and rigorously? 

Reviewer #1: I Don't Know

Reviewer #2: Yes

3. Have the authors made all data underlying the findings in their manuscript fully available?

Reviewer #1: No

Reviewer #2: Yes

4. Is the manuscript presented in an intelligible fashion and written in standard English?

Reviewer #1: Yes

Reviewer #2: Yes

5. Review Comments to the Author

Reviewer #1: The paper aims to assess cost-effectiveness of the target-D intervention against usual care. Mental health and primary care are both very important subjects for public health. Evaluation of e-health is also of prime importance. The work is original but more methodological precisions were needed.

Main concerns:

1) A micro-costing approach was used to estimate the cost of the target-D intervention except for the clinician-guided iCBT program (moderate prognostic group) (page 5, lines 112-113: “The average cost between two Australia clinician-guides iCBT programs was applied to participants in the moderate prognostic group”). Why?

2) Some costs appeared to be research-induced (e.g. p5, lines 105-106 “personnel time to approach individuals in the GP waiting room involved one minute per encounter”). Please clarify

3) The drop-out rate was very important for the RUQ data and the AQoL-8D data. Not enough information is given to the lector on how the missing data were treated. Bootstrapped data need to be stratified by treatment arm. The process of missing data need to be tested, not assumed (page 6, line 140: “assuming data were missing at random”). Tests should be presented and discussed. If data were not MAR, extensive sensitivity analyses (scenarios) on costs and quality-of-life utilities need to be conducted (not only complete case analysis, valid if and only if missing data were MCAR). Please give reference if this issue was treated in a previous paper.

4) Two GLM had been estimated, one for costs (link=log; family=gamma) and another for utility scores (link=identity; family=Gaussian). In the paper, it is not clear if these models were re-estimated for each bootstrapped sample or only once. If ICER is computed from estimated coefficients, how the ratio was converted into a difference for costs? How potential correlation between errors terms of the QALY equation and the cost equation were taken into account in the analysis?

5) Concerning the QALY equation, was the value at baseline systematically included among covariates? (not clear page 6, lines 147-149: “All GLM models were estimated with and without adjustment for several baselines specified in the study protocol- i.e. baseline PHQ-9 score (not QALY, as requested in guidelines),general practice and prognostic group”)

6) It could be interesting to present details on cost provided by microcosting for each level of intervention. Page 8, lines 191-194, be more affirmative “This was likely due to the high-cost nature of collaborative care delivered to participants in the severe group”.

7) Acceptability curves could be estimated for the 3 prognostic groups.

8) The conclusion (page 19, line 338-341) was very strong, not really in line with methodological issues mentioned page 18, lines 318-319 and 325-327

Reviewer #2: The authors present an economic evaluation of the Target-D intervention, based on resource utilization information collected during a clinical trial of Target-D versus usual care in Melbourne, Australia. Results are presented both from a health sector perspective and a societal perspective. Authors conclude that Target-D likely has good value for health care decision makers. The manuscript is well written. I only have a few minor recommendations for the authors.

1. line 43: authors state that health sector and societal costs were "comparable" between trial arms at 3 and 12 months. Authors should replace "comparable" with "not significantly different" since authors did not do a specific test for equality of the costs.

2. Authors should provide the number of control and intervention participants in each of the prognostic groups (minimal/mild, moderate, severe), rather than relying on readers to go to the published paper on trial results to get this information. This can likely just be put in the text in lines 177-178.

3. lines 208 (note under Table 1), 217 (note under Table 2), and 235 (note under Table 3): "partcipants" should be "participants"

6. PLOS authors have the option to publish the peer review history of their article (what does this mean?). If published, this will include your full peer review and any attached files.

Reviewer #1: No

Reviewer #2: No

---

## [Author Response · Author response to Decision Letter 0]

7 Feb 2022

Editor #1:

E1.1 – 1. Please ensure that your manuscript meets PLOS ONE's style requirements, including those for file naming. The PLOS ONE style templates can be found at 

and

We have sought to comply with the PLOS ONE style templates the best we can.

E1.2 – 2. Please update your submission to use the PLOS LaTeX template. The template and more information on our requirements for LaTeX submissions can be found at 

http://journals.plos.org/plosone/s/latex.

For now, we have opted to submit our manuscript using Word document files. We will be more than willing to provide a submission in the PLOS LaTeX template following the acceptance of our paper.

E1.3 – 3. Thank you for stating the following in the Acknowledgments Section of your manuscript: “The authors would like to thank all the patients, family physicians, and clinics who took part in Target-D ... Finally, we thank staff at the former Melbourne Networked Society Institute (MNSI) who were funded to build the Target-D website.” We note that you have provided funding information that is not currently declared in your Funding Statement. However, funding information should not appear in the Acknowledgments section or other areas of your manuscript. We will only publish funding information present in the Funding Statement section of the online submission form. Please remove any funding-related text from the manuscript and let us know how you would like to update your Funding Statement. Please include your amended statements within your cover letter; we will change the online submission form on your behalf.

We have modified the acknowledgements to remove all references to funding information. Please see the amended text below.

The authors would like to thank all the patients, family physicians, and clinics who took part in Target-D; and the many research assistants who assisted with data collection. The data used to develop the clinical prediction tool were collected as a part of the diamond project (NHMRC project ID: 299869, 454463, 566511 and 1002908). We acknowledge the 30 dedicated family physicians, their patients, and clinic staff for making the diamond study possible. We also acknowledge staff and students at the School of Computing and Information Systems at the University of Melbourne for early work that informed the presentation of the e-health platform as well as the focus group participants that provided feedback on early versions of the Target-D materials. Finally, we thank staff at the former Melbourne Networked Society Institute (MNSI) who built the Target-D website.

E1.4 – 4. In your Data Availability statement, you have not specified where the minimal data set underlying the results described in your manuscript can be found. PLOS defines a study's minimal data set as the underlying data used to reach the conclusions drawn in the manuscript and any additional data required to replicate the reported study findings in their entirety. All PLOS journals require that the minimal data set be made fully available. For more information about our data policy, please see http://journals.plos.org/plosone/s/data-availability. Upon re-submitting your revised manuscript, please upload your study’s minimal underlying data set as either Supporting Information files or to a stable, public repository and include the relevant URLs, DOIs, or accession numbers within your revised cover letter. For a list of acceptable repositories, please see http://journals.plos.org/plosone/s/data-availability#loc-recommended-repositories. Any potentially identifying patient information must be fully anonymized. Important: If there are ethical or legal restrictions to sharing your data publicly, please explain these restrictions in detail. Please see our guidelines for more information on what we consider unacceptable restrictions to publicly sharing data: http://journals.plos.org/plosone/s/data-availability#loc-unacceptable-data-access-restrictions. Note that it is not acceptable for the authors to be the sole named individuals responsible for ensuring data access. We will update your Data Availability statement to reflect the information you provide in your cover letter.

Ethics approval for the clinical trial underlying the submitted manuscript was obtained through the University of Melbourne. We have contacted Ms Hilary Young, Secretary for the Medicine and Dentistry Human Ethics Sub-Committee (HESC) at the University of Melbourne, to provide us with guidance on this matter (Phone: +61 3 8344 8595, Email: hilary.young@unimelb.edu.au). We have attached a copy of the resulting correspondence. To summarise, we have been notified that the plain language statement of the original study does not provide a contingency for study participants to provide informed consent for any prospective data sharing, particularly given that collected data is potentially identifying and involves sensitive personal information. As such, we are unable to release the data as part of any minimum dataset. We have amended our Data Availability Statement to read:

Ethical restrictions prevent the sharing of potentially sensitive data provided by study participants over the course of the Target-D clinical trial (Australian New Zealand Clinical Trials Registry ACTRN12616000537459). No contingency was included in the original plain language statement for study participants to provide informed consent to any prospective sharing of their personal data, whether as part of a minimum dataset or another form. For all enquiries regarding the Target-D clinical trial and the underlying dataset, please contact the chief investigator, Prof Jane Gunn (j.gunn@unimelb.edu.au). For general enquiries regarding the ethics approval of the trial, please contact the Office of Research Ethics and Integrity (OREI) at The University of Melbourne (HumanEthics-Enquiries@unimelb.edu.au).

As a compromise to our inability to provide a minimum dataset, we have now included an additional supplementary appendix (S2 Appendix) that contains summary metadata on all in-scope data variables, alongside a copy of the Stata do-file. The following sentence has been added to the end of the ‘Statistical analysis’ section of the Methods in lines 175-176:

A summary list of all variables included in the statistical analysis is provided in S2 Appendix, alongside the Stata do-file used to implement the statistical analysis.

 

Reviewer #1:

The paper aims to assess cost-effectiveness of the target-D intervention against usual care. Mental health and primary care are both very important subjects for public health. Evaluation of e-health is also of prime importance. The work is original but more methodological precisions were needed.

Main concerns:

R1.1 – 1) A micro-costing approach was used to estimate the cost of the target-D intervention except for the clinician-guided iCBT program (moderate prognostic group) (page 5, lines 112-113: “The average cost between two Australia clinician-guides iCBT programs was applied to participants in the moderate prognostic group”). Why?

There is no definitive, gold standard unit cost for clinician-guided iCBT in Australia. Instead, we have available two alternative unit costs that provide a low-to-high range of possible values. The point value used in the base case analysis comprised the average of the low and high values. This is the rationale for why we performed a subsequent sensitivity analysis analysing the impact of adopting the highest unit cost ($222 per person). In response to this comment, we have added the following sentence to improve clarity on lines 116-117:

In this instance, the two programs represent a low-to-high range of possible unit cost values for clinician-guided iCBT in Australia. A subsequent sensitivity analysis was done to test the impact of using the highest unit cost, rather than the average. 

R1.2 – 2) Some costs appeared to be research-induced (e.g. p5, lines 105-106 “personnel time to approach individuals in the GP waiting room involved one minute per encounter”). Please clarify

We made sure to include costs that would occur in routine practice and to exclude research-related costs. Resource use involving the research assistants (e.g., approaching individuals in the GP waiting room or periodic check-in phone calls) will need to be performed by similarly qualified staff if the intervention were to be implemented as part of routine practice. We have added a sentence to lines 120-122 to clarify this point:

It is anticipated that all costed activities described above that involve research assistants will likely require similarly qualified staff to facilitate the implementation of the intervention as part of routine practice.

R1.3 – 3) The drop-out rate was very important for the RUQ data and the AQoL-8D data. Not enough information is given to the lector on how the missing data were treated. Bootstrapped data need to be stratified by treatment arm. The process of missing data need to be tested, not assumed (page 6, line 140: “assuming data were missing at random”). Tests should be presented and discussed. If data were not MAR, extensive sensitivity analyses (scenarios) on costs and quality-of-life utilities need to be conducted (not only complete case analysis, valid if and only if missing data were MCAR). Please give reference if this issue was treated in a previous paper.

We thank the reviewer for encouraging us to be clear in how we have chosen to address the problem of missing data. In response to this comment, we have now added a new section to the supplementary materials, ‘Supplementary Text S5. Analysis of missing data mechanisms’. In this supplement, we have provided a detailed exploration of missing data patterns and an empirical rationale for why we have concluded that there is sufficient evidence to infer that missing utility/cost data can be considered missing at random (as opposed to missing not at random). Based on these analyses, we identified several baseline sociodemographic variables that were associated with the likelihood of missing utility/cost values. We have consequently included these variables as adjustment covariates in the multiple imputation analysis, which was also updated. All results based on these methodological refinements have been amended accordingly.

We have modified text on lines 144-150 to read:

Multiple imputation methods were implemented in Stata to account for missing data that were deemed missing at random following several exploratory analyses presented in Supplementary Text 5 in S1 Appendix. Missing cost and outcomes data were imputed 100 times using multiple imputation by chained equations (MICE), with predictive mean matching and adjustment for baseline covariates associated with data missingness – i.e., trial arm, clinic, age, gender, highest level of education and having visited a psychologist/counsellor in the past 12 months. 

R1.4 – 4) Two GLM had been estimated, one for costs (link=log; family=gamma) and another for utility scores (link=identity; family=Gaussian). In the paper, it is not clear if these models were re-estimated for each bootstrapped sample or only once. If ICER is computed from estimated coefficients, how the ratio was converted into a difference for costs? How potential correlation between errors terms of the QALY equation and the cost equation were taken into account in the analysis?

The reviewer is justified in their call for further descriptive detail on the methods used to implement bootstrapping. We confirm that the ICER was computed using estimated GLM coefficients of the difference in mean costs and the difference in mean QALYs. In the original analysis, we adopted a resampling method that encompassed, ‘bootstrapping nested in multiple imputation’. Since then, we have encountered recommendations by Brand et al., 2019 (doi: 10.1002/sim.7956) and Prof Andy Briggs who collectively advocate, ‘single imputation nested in bootstrapping’. Based on these recommendations, we have revised our analytic approach and modified our description of the methods/results accordingly.

The text on lines 161-173 has now been amended to read:

Incremental cost-effectiveness ratios (ICERs) were calculated as the difference in mean costs between the intervention and control arms divided by the difference in mean QALYs. ICERs were calculated by study perspective (health sector and societal), follow-up period (3 and 12 months) and, for the subgroup analysis, by prognostic group (total, minimal/mild, moderate and severe). A resampling method comprising single imputation nested in bootstrapping [17] was used to quantify the impact of input parameter uncertainty around the resulting differences in mean costs/QALYs and the mean ICERs. This method works by generating a single call to the MICE procedure to produce a complete dataset with which to analyse GLMs of costs/QALYs within each bootstrap resample. Following the generation of 1,000 bootstrap resamples, the bootstrap percentile method was used to estimate 95% confidence intervals (95% CI) around the differences in mean costs/QALYs and the mean ICERs [18]. The intervention was considered cost-effective if the resulting ICER was less than the Australian willingness-to-pay threshold of A$50,000 per QALY [19-21].

Furthermore, we have now included an additional supplementary appendix (S2 Appendix) that contains both the Stata do-file and a summary list of data variables. The following sentence has been added to the end of the ‘Statistical analysis’ section of the Methods in lines 175-176:

A summary list of all variables included in the statistical analysis is provided in S2 Appendix, alongside the Stata do-file used to implement the statistical analysis.

R1.5 – 5) Concerning the QALY equation, was the value at baseline systematically included among covariates? (not clear page 6, lines 147-149: “All GLM models were estimated with and without adjustment for several baselines specified in the study protocol- i.e. baseline PHQ-9 score (not QALY, as requested in guidelines),general practice and prognostic group”)

The reviewer has made an important critique of our methods. Our initial analysis did not adjust for baseline utility scores derived using the AQoL-8D measure as we were narrowly focussed on reproducing the primary outcomes analysis, which made adjustments for the baseline PHQ-9 score. Moreover, we had a priori postulated that baseline PHQ-9 scores would be (in theory) highly correlated to baseline AQoL-8D scores. Following the reviewer’s comment, we have made a decision to re-analyse QALY outcomes after making an additional adjustment for baseline AQoL-8D scores.

In the methods, we have amended lines 155-159:

All GLMs were estimated with and without adjustment for several baseline covariates specified in the study protocol ‒ i.e., baseline PHQ-9 score, general practice and prognostic group [7]. Baseline AQoL-8D scores were also included as an additional baseline covariate for GLMs involving QALY outcomes.

The results that are reported in Table 3 now reflect these changes. Additionally, the first footnote to the results presented in Table 3 on line 253 has also been amended to reflect the addition of the baseline AQoL-8D score as a baseline covariate. 

R1.6 – 6) It could be interesting to present details on cost provided by microcosting for each level of intervention. Page 8, lines 191-194, be more affirmative “This was likely due to the high-cost nature of collaborative care delivered to participants in the severe group”.

Detailed costs encompassing the microcosting approach are presented for each intervention level in Supplementary Table S3. We have amended to sentence on lines 212-214 to read:

This was likely due to the high-cost nature of collaborative care delivered to participants in the severe group (see Supplementary Table 3 in S1 Appendix for detailed costs).

R1.7 – 7) Acceptability curves could be estimated for the 3 prognostic groups.

We appreciate the reviewer’s suggestion here. However, we have opted not to present in-depth results for the three prognostic groups (i.e., cost-effectiveness planes or cost-effectiveness acceptability curves) given that these are subgroup analyses that are underpowered to detect statistically significant differences, particularly when compared to the aggregate findings. We have amended text in lines 162-165 to emphasise the fact that the analysis of prognostic groups encompasses a subgroup analysis:

ICERs were calculated by study perspective (health sector and societal), follow-up period (3 and 12 months) and, for the subgroup analysis, by prognostic group (total, minimal/mild, moderate and severe).

R1.8 – 8) The conclusion (page 19, line 338-341) was very strong, not really in line with methodological issues mentioned page 18, lines 318-319 and 325-327

We have amended the relevant texts on lines 359-362 and lines 365-366 to soften conclusions drawn based on our study findings. These texts now read as follows:

The results of this study suggest that stepped care may be dominant when compared to usual care. Additionally, study findings appear to support the existing literature by suggesting that stepped care for depression can deliver improved clinical outcomes without increasing costs.

The Target-D intervention is likely to represent good value for money and provides indicative support for further development of digitally supported mental health care.

 

Reviewer #2:

The authors present an economic evaluation of the Target-D intervention, based on resource utilization information collected during a clinical trial of Target-D versus usual care in Melbourne, Australia. Results are presented both from a health sector perspective and a societal perspective. Authors conclude that Target-D likely has good value for health care decision makers. The manuscript is well written. I only have a few minor recommendations for the authors.

R2.1 – 1. line 43: authors state that health sector and societal costs were "comparable" between trial arms at 3 and 12 months. Authors should replace "comparable" with "not significantly different" since authors did not do a specific test for equality of the costs.

We thank the reviewer for this suggestion and have amended the Abstract text accordingly (see line 43).

R2.2 – 2. Authors should provide the number of control and intervention participants in each of the prognostic groups (minimal/mild, moderate, severe), rather than relying on readers to go to the published paper on trial results to get this information. This can likely just be put in the text in lines 177-178.

We have added this information to lines 194-198 at the beginning of the Results section, as requested by the reviewer.

R2.3 – 3. lines 208 (note under Table 1), 217 (note under Table 2), and 235 (note under Table 3): "partcipants" should be "participants"

We thank the reviewer for spotting this mistake. The spelling of this word has now been corrected in all relevant locations.

---

## [Decision Letter · Decision Letter 1]

7 Mar 2022

PONE-D-21-25251R1Economic evaluation of the Target-D platform to match depression management to severity prognosis in primary care: a within-trial cost-utility analysisPLOS ONE

Dear Dr. Lee,

Thank you for submitting your manuscript to PLOS ONE. After careful consideration, we feel that it has merit but does not fully meet PLOS ONE’s publication criteria as it currently stands. Therefore, we invite you to submit a revised version of the manuscript that addresses the 2 minor points raised during the review process:

1) As the effectiveness difference between arms is very small (non significant), I should be preferable to estimate the 95% CI for ICER using the Fieller's method instead of the bootstrap method. The former is less sensitive to misinterpretation of the CI bounds than the latter [see https://www.iresp.net/wp-content/uploads/2018/12/Siani-article-3.pdf] 2) In the QALY equation, the utility score at inclusion should be included as covariate (not baseline AQoL-8D score) [see Willan and Briggs, Statistical analysis of cost-effectiveness data, Statistics in Practice, Wiley, page 24-25]

We look forward to receiving your revised manuscript.

Kind regards,

Isabelle Durand-Zaleski

Academic Editor

PLOS ONE

Journal Requirements:

Additional Editor Comments (if provided):

You have addressed most of the reviewers' comments, I recommend that you take into account the 2 suggestions of reviewer 1 in your final version.

1) As the effectiveness difference between arms is very small (non significant), I should be preferable to estimate the 95% CI for ICER using the Fieller's method instead of the bootstrap method. The former is less sensitive to misinterpretation of the CI bounds than the latter [see https://www.iresp.net/wp-content/uploads/2018/12/Siani-article-3.pdf]

2) In the QALY equation, the utility score at inclusion should be included as covariate (not baseline AQoL-8D score) [see Willan and Briggs, Statistical analysis of cost-effectiveness data, Statistics in Practice, Wiley, page 24-25]

Reviewers' comments:

Reviewer's Responses to Questions

**Comments to the Author**

1. If the authors have adequately addressed your comments raised in a previous round of review and you feel that this manuscript is now acceptable for publication, you may indicate that here to bypass the “Comments to the Author” section, enter your conflict of interest statement in the “Confidential to Editor” section, and submit your "Accept" recommendation.

Reviewer #1: All comments have been addressed

Reviewer #2: All comments have been addressed

2. Is the manuscript technically sound, and do the data support the conclusions?

Reviewer #1: Yes

Reviewer #2: (No Response)

3. Has the statistical analysis been performed appropriately and rigorously? 

Reviewer #1: Yes

Reviewer #2: (No Response)

4. Have the authors made all data underlying the findings in their manuscript fully available?

Reviewer #1: No

Reviewer #2: (No Response)

5. Is the manuscript presented in an intelligible fashion and written in standard English?

Reviewer #1: Yes

Reviewer #2: (No Response)

6. Review Comments to the Author

Reviewer #1: Thank you to the authors for adressing all my comments on the previous version of the paper.

I have two (marginal) left comments :

1) As the effectiveness difference between arms is very small (non significant), I should be preferable to estimate the 95% CI for ICER using the Fieller's method instead of the bootstrap method. The former is less sensitive to misinterpretation of the CI bounds than the latter [see https://www.iresp.net/wp-content/uploads/2018/12/Siani-article-3.pdf]

2) In the QALY equation, the utility score at inclusion should be included as covariate (not baseline AQoL-8D score) [see Willan and Briggs, Statistical analysis of cost-effectiveness data, Statistics in Practice, Wiley, page 24-25]

Reviewer #2: (No Response)

7. PLOS authors have the option to publish the peer review history of their article (what does this mean?). If published, this will include your full peer review and any attached files.

Reviewer #1: No

Reviewer #2: No

---

## [Author Response · Author response to Decision Letter 1]

21 Mar 2022

Reviewer #1:

Thank you to the authors for adressing all my comments on the previous version of the paper. I have two (marginal) left comments :

R1.1 – 1) As the effectiveness difference between arms is very small (non significant), I should be preferable to estimate the 95% CI for ICER using the Fieller's method instead of the bootstrap method. The former is less sensitive to misinterpretation of the CI bounds than the latter [ see https://www.iresp.net/wp-content/uploads/2018/12/Siani-article-3.pdf ]

We thank the reviewer for suggesting Fieller's theorem as a potentially more accurate method by which to estimate the 95% confidence interval for the mean ICER; under the scenario when the expected value of the difference in effectiveness outcomes between trial arms approaches zero (as occurs in our study). We have read through the article by Siani et al. (2003) and note that: (1) Fieller's theorem has the potential to produce more accurate 95% confidence interval bounds with excellent coverage over the 95% confidence region - based largely on simulations analysed in the quoted study; and (2) 95% CIs produced using the non-parametric bootstrap percentile method can potentially lead to confidence bounds with a marginally higher coverage of the target 95% confidence region (i.e., 97% coverage of the mean ICER). We conducted a quick search of the literature and struggled to find studies that have replicated the findings of Siani et al., (2003). This makes it difficult to confirm the veracity of the phenomena identified by Siani et al. (2003). Even so, we concede that the potential for the bias identified by Siani et al. (2003) remains.

If the aforementioned bias were to transpire, then we contend that such imprecision in the estimation of 95% confidence bounds will not have a material impact on the interpretation of our study findings. This is due largely to the bootstrap resampling results which indicated a high degree of uncertainty around the expected value of mean ICERs estimated across all base case and subgroup analyses (i.e., bootstrap resamples for mean ICERs consistently covered all four quadrants of the cost-effectiveness plane). As such, the 95% confidence bounds produced by the bootstrap percentile method did not approach the nominated WTP threshold of A$50,000 per QALY. This was especially true when the lower and upper 95% confidence bounds spanned the South-East ('dominant') and North-West ('dominated') quadrants of the cost-effectiveness plane. In summary, any (comparatively small) imprecision around the 95% confidence bounds presented in Table 4 is expected to be inconsequential when compared to the extreme range of lower and upper 95% confidence bounds.

In response to this comment, we have added a note to Table 4 stating that:

The mean difference in QALYs between trial arms was observed to approach zero across all base case and subgroup analyses. This can potentially lead to the lower and upper bounds of a 95% confidence interval, derived using the bootstrap percentile method, encompassing a marginally higher coverage than the target 95% confidence region (e.g., 97% coverage of the mean ICER) [22]. Even so, any resulting imprecision in the estimation of the 95% confidence bounds will likely be inconsequential to the interpretation of study findings given the wide range of ICER values that were consistently observed between the lower and upper confidence bounds (e.g., confidence bounds ranging between 'dominant' and 'dominated'). This reflects the high degree of uncertainty observed across mean ICER values for all base case and subgroup analyses; with bootstrap resamples consistently spanning all four quadrants of the cost-effectiveness plane.

R1.2 – 2) In the QALY equation, the utility score at inclusion should be included as covariate (not baseline AQoL-8D score) [see Willan and Briggs, Statistical analysis of cost-effectiveness data, Statistics in Practice, Wiley, page 24-25]

We apologise to the reviewer for using imprecise terminology that has, in turn, led to this instance of semantic confusion. When we used the term 'baseline AQoL-8D score', our intended meaning was 'baseline AQoL-8D utility weight' – i.e., utility weights estimated based on scoring the AQoL-8D multi-attribute utility instrument. In response to this comment, we have changed all instances of 'AQoL-8D score(s)' to 'AQoL-8D utility weight(s)'.

---

## [Editor Report · Decision Letter 2]

12 May 2022

Economic evaluation of the Target-D platform to match depression management to severity prognosis in primary care: a within-trial cost-utility analysis

PONE-D-21-25251R2

Dear Dr. Lee,

We’re pleased to inform you that your manuscript has been judged scientifically suitable for publication and will be formally accepted for publication once it meets all outstanding technical requirements.

Kind regards,

Isabelle Durand-Zaleski

Academic Editor

PLOS ONE
---

## [Editor Report · Acceptance letter]

16 May 2022

PONE-D-21-25251R2 

Economic evaluation of the Target-D platform to match depression management to severity prognosis in primary care: a within-trial cost-utility analysis 

Dear Dr. Lee:

I'm pleased to inform you that your manuscript has been deemed suitable for publication in PLOS ONE. Congratulations! Your manuscript is now with our production department. 

Kind regards, 

on behalf of

Dr. Isabelle Durand-Zaleski 

Academic Editor

PLOS ONE